

# Intercomparison and evaluation of satellite peroxyacetyl nitrate observations in the upper troposphere - lower stratosphere

R. J. Pope[1,2], N. A. D. Richards[1,2], M. P. Chipperfield[1,2], D. P. Moore[3,4], S. A. Monks[6,7], S. R. Arnold[1], N. Glatthor[5], M. Kiefer[5], T. J. Breider[8], J. J. Harrison[3,4], J. J. Remedios[3,4], C. Warneke[6,7], J. M. Roberts[6], G. S. Diskin[9], L. G. Huey[10], A. Wisthaler[11,12], E. C. Apel[13], P. F. Bernath[14], and W. Feng[1,15]

[1]School of Earth and Environment, University of Leeds, Leeds, UK
[2]National Centre for Earth Observation, University of Leeds, Leeds, UK
[3]Department of Physics and Astronomy, University of Leicester, Leicester, UK
[4]National Centre for Earth Observation, University of Leicester, Leicester, UK
[5]Karlsruhe Institute of Technology, Institute of Meteorology and Climate Research, Karlsruhe, Germany
[6]Earth System Research Laboratory, National Oceanic and Atmospheric Administration, Boulder, CO, USA
[7]Cooperative Institute for Research in Environmental Sciences, University of Colorado, Boulder, CO, USA
[8]School of Engineering and Applied Sciences, Harvard University, Cambridge, USA
[9]NASA Langley Research Center, Chemistry and Dynamics Branch, Hampton, VA, USA
[10]Georgia Institute of Technology, Atlanta, GA, USA
[11]University of Innsbruck, Innsbruck, Austria
[12]University of Oslo, Oslo, Norway





[13]Atmospheric Chemistry Division, National Centre for Atmospheric Research, Boulder, CO, USA
[14]Department of Chemistry and Biochemistry, Old Dominion University, Norfolk, VA, USA
[15]National Centre for Atmospheric Science, University of Leeds, Leeds, UK

Correspondence to: Richard Pope
(r.j.pope@leeds.ac.uk)



# Abstract

Peroxyacetyl nitrate (PAN) is an important chemical species in the troposphere as it aids the long-range transport of $NO_x$ and subsequent formation of $O_3$ in relatively clean remote regions. Over the past few decades observations from aircraft campaigns and surface sites have been used to better understand the regional distribution of PAN. However, recent measurements made by satellites allow for a global assessment of PAN in the upper troposphere - lower stratosphere (UTLS). In this study, we investigate global PAN distributions from two independent retrieval methodologies, based on measurements from the Michelson Interferometer for Passive Atmospheric Sounding (MIPAS) instrument, on board ENVISAT from the Institute of Meteorology and Climate Research (IMK), Karlsruhe Institute of Technology and the Department of Physics and Astronomy, University of Leicester (UoL). Retrieving PAN from MIPAS is challenging due to the weak signal in the measurements and contamination from other species. Therefore, we compare the two MIPAS datasets with observations from the Atmospheric Chemistry Experiment Fourier Transform Spectrometer (ACE-FTS), in-situ aircraft data and the TOMCAT 3-D chemical transport model. MIPAS shows peak UTLS PAN concentrations over the biomass burning regions (e.g. ranging from 150 to >200 pptv at 150 hPa) and during the summertime Asian monsoon as enhanced convection aids the vertical transport of PAN from the lower atmosphere. At 150 hPa, we find significant differences between the two MIPAS datasets in the tropics, where IMK PAN concentrations are larger by 50-100 pptv. Comparisons between MIPAS and ACE-FTS show better agreement with the UoL MIPAS PAN concentrations at 200 hPa, but with mixed results above this altitude. TOMCAT generally captures the magnitude and structure of climatological aircraft PAN profiles within the observational variability allowing it to be used to investigate the MIPAS PAN differences. TOMCAT-MIPAS comparisons show that the model is both positively (UoL) and negatively (IMK) biased against the satellite products. These results show that satellite PAN observations are able to detect realistic spatial variations in PAN in the UTLS, but further work is needed



to resolve differences in existing retrievals to allow quantitative use of the products.

# 1 Introduction

Peroxyacetyl nitrate (PAN) is a key species in the chemistry of the troposphere. PAN is produced in polluted regions through the reaction of hydrocarbons, which contain an acetyl group ($-C(O)CH_3$) such as acetone and acetaldehyde, with OH and $O_2$ to form the peroxyacetyl radical ($CH_3C(O)OO$). The subsequent reversible temperature dependent reaction of the peroxyacetyl radical with nitrogen dioxide ($NO_2$) produces PAN,

$$CH_3C(O)OO + NO_2 + M \rightleftharpoons PAN + M \qquad [R1]$$

where M is a third body. PAN produced at the surface can be uplifted into the cold upper troposphere (UT) where it has a relatively long lifetime of several months (Singh, 1987; Talukdar et al., 1995) enabling it to be transported over large distances. PAN therefore acts as a reservoir for $NO_x$ (NO + $NO_2$) in the UT. When UT air masses descend and warm, PAN breaks down to release $NO_2$, which may promote ozone production in regions with small local $NO_x$ sources (Wang et al., 1998; Hudman et al., 2004). PAN therefore plays an important role in the long-range transport of pollution to remote clean areas and has a strong influence on the oxidizing capacity of the troposphere.

Initial observations of tropospheric PAN came from a small number of aircraft campaigns (Singh et al., 1996, 2000; Russo et al., 2003; Roberts et al., 2004; Lewis et al., 2007). These observations showed that PAN is widespread throughout the mid and upper troposphere, with remote background concentrations of between 50 and 100 pptv (Singh et al., 2000) rising to over 600 pptv in polluted air masses (Russo et al., 2003). The first global measurements of upper tropospheric PAN were retrieved from the Michelson Interferometer for Passive Atmospheric Sounding (MIPAS) instrument on board ENVISAT (Glatthor et al., 2007; Moore and Remedios, 2010; Wiegele et al.,



2012). Glatthor et al. (2007) demonstrated the ability of MIPAS to retrieve PAN in the upper troposphere/lower stratosphere (UTLS) region with a height resolution of 3.5-6 km. They also demonstrated that MIPAS was able to observe PAN in southern hemisphere biomass burning plumes, with similar concentrations to previous aircraft campaigns. Furthermore, Moore and Remedios (2010) demonstrated that MIPAS is able to capture the seasonal cycle of PAN in the UTLS. In the BORTAS (quantifying the impact of BOReal forest fires on Tropospheric oxidants over the Atlantic using Aircraft and Satellites) campaign, Tereszchuk et al. (2013) showed that three PAN profiles from ACE-FTS (Atmospheric Chemistry Experiment Fourier Transform Spectrometer) agree with MIPAS data (from Moore and Remedios (2010)) within the respective error margins above 150 hPa when observing the biomass burning outflow from North America in July 2011.

Previous modelling studies of PAN have concentrated on the effects of volatile organic compounds (VOCs) on PAN formation. Pfister et al. (2008) showed that isoprene can contribute up to 29% of the annual global atmospheric PAN burden. Ito et al. (2007) found a 40% increase in modelled PAN concentrations with the inclusion of aromatic and terpenoid hydrocarbons and hydroxyl alkyl nitrates produced from isoprene. Fischer et al. (2014) demonstrated that acetaldehyde (44%) and methyglyoxal (37%) are the primary VOCs leading to the formation of the peroxyacetyl radical. Isoprene (37%) and alkanes (14%) are the main emissions aiding PAN formation. Emmons et al. (2015) led a model inter-comparison project (including models such as TOMCAT and GEOS-Chem) looking at tropospheric chemistry in the Arctic. They found that the majority of models reproduce the same seasonal cycle at 700 hPa between 50-70°N, with peak PAN in March-May. When compared with flight campaigns, the majority of the models (including TOMCAT) overestimated PAN concentrations in the lower troposphere. Arnold et al. (2015) investigated the influence of biomass burning on northern high latitude tropospheric PAN, and found that chemical transport models (CTMs) using ERA-Interim meteorology overestimated the PAN observations, while those that used GEOS-5 meteorology underestimated PAN. Fadnavis et al. (2014) used the ECHAM5-



HAMMOZ global chemistry-climate model (CCM) and MIPAS PAN observations (from Karlsruhe Institute for Technology) between 2002-2011 to detect peak vertical transport of PAN into the UTLS during the Asian summertime monsoon. However, compared to MIPAS, ECHAM5-HAMMOZ had a low bias in the seasonal cycle (approximately 100 pptv) in June-August.

In this paper we use a synthesis of satellite and aircraft data and the TOMCAT CTM to find robust features of PAN in the UTLS and quantify its uncertainty. In particular, we compare two different retrievals of PAN from the MIPAS satellite with ACE-FTS retrievals. Section 2 describes the observations used and the TOMCAT model configuration. We discuss our model-observation inter-comparisons in Section 3 and present our conclusions in Section 4.

## 2   Observations and Model

### 2.1   Satellite and Aircraft Observations

The primary observations used in this paper are retrieved from the MIPAS and ACE-FTS satellite instruments. We used data from two different MIPAS retrievals performed by the University of Leicester (UoL) and the Institute of Meteorology and Climate Research, Karlsruhe Institute for Technology (IMK). The UoL and IMK MIPAS PAN retrieval methods are discussed by Moore and Remedios (2010) and Glatthor et al. (2007), respectively. We investigate the PAN data between the two retrieval processes and then also compare with observations from ACE-FTS.

MIPAS flew on-board the ESA ENVISAT satellite and obtained data between 2002-2012. It was a limb viewing instrument with an orbit height of 800 km, which measured atmospheric gases in the mid-IR (685-2410 cm$^{-1}$; Fischer et al. (2008)). It had a descending equatorial local time crossing (LT) of 10.00 am and approximately 14.4 orbits per day resulting in approximately 1400 profiles. The measurements, in reduced resolution nominal mode, had 27 tangential altitudes per limb scan. The lowermost



(uppermost) tangent altitudes ranged approximately from 5 km (70 km) near the poles to 12 km (77 km) at the equator (Wiegele et al., 2012).

The ACE-FTS instrument, onboard the SCISAT satellite, is a limb-viewing instrument using solar occultation to measure atmospheric spectra over the IR region 750-4400 cm$^{-1}$ continuously at high spectral resolution (0.02 cm$^{-1}$). It can make measurements from 5 km to 150 km in altitude with a field-of-view of about 3 km and can record up to 30 occultations (sunrise and sunset) per day (Bernath et al. (2005); Tereszchuk et al. (2013)). Therefore, the spatial coverage of ACE-FTS is less than that of MIPAS but it has similar vertical resolution.

We use aircraft measurements of PAN and its precursors to assess the skill of the model in the lower atmosphere where MIPAS does not retrieve PAN. We use aircraft data from the Aerosol, Radiation, and Cloud Processes affecting Arctic Climate (ARC-PAC - (Brock et al., 2011)) project (31/03/2008 - 19/04/2008, 22/06/2008 - 12/07/2008) and the Arctic Research of the Composition of the Troposphere from Aircraft and Satellites (ARCTAS - (Jacob et al., 2010)) mission (A, B and CARB; 01/04/2008 - 19/04/2008, 18/06/2008 -13/07/2008) over North America and Greenland to compare directly to TOMCAT in time and space. The flight tracks of the campaigns are shown in Figure 1. We also compare TOMCAT with aircraft measurements of PAN from the multi-year regional aircraft composite dataset compiled by Emmons et al. (2000).

## 2.2 TOMCAT 3-D Model

In this study we use the TOMCAT three-dimensional (3-D) off-line CTM (e.g. Chipperfield et al. (1993); Stockwell and Chipperfield (1999); Chipperfield (2006)). The model is forced using winds, temperature, and humidity from European Centre for Medium-Range Weather Forecasts (ERA-Interim) meteorological analyses. The standard model uses 82 advected tracers and 229 gas-phase reactions (Emmons et al., 2015), which includes the extended tropospheric chemistry (ExTC) scheme used by Richards et al. (2013).The VOC degradation chemistry scheme incorporates the oxidation of monoterpenes, C2-C4 alkanes, toluene, ethene, propene, acetone, methanol



and acetaldehyde, which was implemented by Monks (2011). The model chemistry scheme includes the Mainz condensed isoprene oxidation mechanism (MIM) (Pöschl et al., 2000). TOMCAT also includes heterogeneous $N_2O_5$ hydrolysis using on-line size-resolved aerosol from the Global Model of Aerosol Processes (GLOMAP) model (Mann et al., 2010). Aerosol types have individual uptake coefficients as parameterized by Evans and Jacob (2005), with the exception of dust which is based on Mogili et al. (2006). Short-lived species (e.g. OH) are not advected and assumed to be in photochemical steady-state. Tracer advection by the resolved winds is performed using the scheme of Prather (1986). Subgrid scale transport is performed using the Tiedtke convection scheme (Tiedtke, 1989; Stockwell and Chipperfield, 1999) and the Holtslag and Boville (1993) parameterization for turbulent mixing in the boundary layer following the method of Wang et al. (1999). Where available, kinetic data are taken from IUPAC (http:www.iupac-kinetic.ch.cam.ac.uk) and for other reactions, we use the Leeds Master Chemical Mechanism (MCM). The model anthropogenic emissions come from the Streets v1.2 inventory (provided by D. Streets (Argonne National Lab)), which is a composite of several regional emissions inventories (Emmons et al., 2015). The MACCity inventory (Granier et al., 2011) is used for the natural emissions and biomass burning emissions come from the Global Fire Emissions Database (GFED) v3.1 inventory (Randerson et al., 2013).The model was initialised at the start of 2006, using a restart (initialisation) file from previous simulations, which resulted in a model spin up period of one year.

In order to compare TOMCAT with MIPAS, the model global fields were sampled at each individual MIPAS profile location and matched in time to the nearest 3 hours. The resulting TOMCAT profiles were then interpolated in the vertical to the retrieved pressure grid so the MIPAS averaging kernels (AK) could be applied. This accounts for the satellite sensitivity to retrieving PAN in the atmosphere and allows for like-for-like comparisons. Both retrieval methods for UoL and IMK have MIPAS AKs with peak sensitivity between approximately 10-15 km (Wiegele et al., 2012; Moore and Remedios, 2010). The UoL (Eq.1) and IMK (Eq. 2) MIPAS AKs are applied as:



$$y = e^{A(\ln x - \ln x_a) + \ln x_a} \tag{1}$$

$$y = A.x \tag{2}$$

where $y$ is the modified TOMCAT PAN retrieval, $A$ is the AK matrix, $x_a$ is the apriori and $x$ is the original model PAN profile. In the IMK retrieval process, the apriori used (Eq. 2) is zero. The UoL AKs are applied to the TOMCAT profiles in log-space because their PAN profiles are retrieved in log-space. Finally, both the TOMCAT and MIPAS profiles were averaged for the two-year time period 2007-2008 onto a horizontal grid of 20° longitude by 10° latitude.

## 3  Results

### 3.1  Satellite PAN Distributions

Figures 2 and 3 show IMK and UoL MIPAS PAN at 150 hPa in December-January-February (DJF), March-April-May (MAM), June-July-August (JJA) and September-October-November (SON) for 2007-2008. The black dashed line represents the dynamical tropopause ($\pm 2$ PVU), based on ERA-Interim potential vorticity data. Figure 4 shows the IMK minus UoL difference in these fields. In all seasons, this dynamical tropopause is at approximately 30°N and 30°S at 150 hPa. The largest PAN concentrations ($>100$ pptv) are typically in the tropical regions (i.e. upper troposphere). The lowest concentrations ($<100$ pptv) tend to be in the lower stratosphere (LS).

For the IMK data (Figure 2) the peak PAN concentrations ($>200$ pptv) occur over Africa in MAM and SON and over southern Asia in JJA. The African peak PAN concentrations are linked to biomass burning and extend from the northern to the southern subtropics in MAM, but are shifted to southern Africa in SON. During SON, large PAN concentrations over the South Atlantic ($>170$ pptv) and along the SH dynamical tropopause (approximately 100 pptv) are linked to outflow from the African biomass





burning plume. In JJA, retrieved PAN concentrations in the UT range between 120-190 pptv and cover the majority of the tropics including Africa, southern Asia and the central Americas. This is probably linked to enhanced tropical upwelling of PAN from the lower troposphere, especially at 20-30°N. Over India, in the summertime Asian monsoon, the dynamical tropopause has propagated northwards (approximately 10°). Therefore, enhanced PAN in the UTLS is observed up to 40°N as shown by Fadnavis et al. (2014), resulting in the largest seasonal concentrations over 200 pptv.

Figure 3 shows that UoL PAN concentrations in the UT tend to be smaller over the tropics and the spatial structures are also not as well defined as the IMK data. In the LS above 40°N and S, UoL MIPAS PAN ranges from 20-70 pptv, which is 0-30 pptv larger than the IMK MIPAS PAN (see Figure 4). At 30-40°N and S in the LS, IMK MIPAS is larger by 0-20 ppbv. Upper tropospheric PAN predominantly ranges between 100-150 pptv, apart from the African biomass burning signals of approximately 160-200 pptv. In JJA, stronger vertical transport from the summertime Asian monsoon results in enhanced UT PAN concentrations (120-150 pptv) in comparisons to other seasons (<100 pptv). However, this summertime Asian monsoon signal in the UoL PAN is not as prominent as in the IMK data.

Figure 4 shows the IMK - UoL MIPAS PAN differences at 150 hPa, where purple polygons indicate regions of significant differences between the two retrievals, which are defined when the mean retrievals $\pm$ their uncertainty ranges do not overlap. The seasonal uncertainty ranges are based on the random and systematic errors in the retrieval process. The random errors reduce with time averaging by a factor of $1/\sqrt{N}$, where $N$ is the number of observations. Systematic errors are not included in the product files so we estimate them from Moore and Remedios (2010) and Glatthor et al. (2007). Moore and Remedios (2010) show that the UoL MIPAS PAN systematic errors range from 10-20% between 350-150 hPa and 40-50% above 150 hPa. Therefore, we assume systematic errors of 20% and 50% at these altitudes, respectively. Glatthor et al. (2007) estimate the IMK MIPAS PAN systematic errors to be approximately 5-20% and 20-30% between 350-150 hPa and above 150 hPa, respectively. Therefore





we assume systematic errors of 20% and 30% in these altitude ranges. For the ACE-FTS retrievals, Tereszchuk et al. (2013) suggest systematic errors of approximately 16%.

Figure 4 shows that in the LS, the IMK PAN concentrations are 0-30 pptv lower with significant differences in regions of the NH high latitudes in MAM and JJA, and the SH high latitudes in DJF. IMK MIPAS PAN tends to be larger in the LS between 30-40°N and S. In the UT, IMK tropical PAN concentrations are significantly larger (50-100 pptv) over northern Africa, South East Asia and in southern Africa. Therefore, the biggest differences are in locations of peak PAN concentrations. However, the IMK-UoL differences are not significant over the equator. In the mid-latitudes, the two MIPAS data sets are in agreement with non-significant differences of -20 to 20 pptv.

To check the IMK and UoL MIPAS PAN differences at other levels, Figures 5 and 6 show the zonal mean IMK and UoL PAN retrievals. The dashed lines again show the location of the dynamical tropopause. Stratospheric PAN concentrations predominantly range between 10-100 pptv in both products. For the IMK PAN the peak zonal mean is 170-220 pptv near the northern mid-latitude tropopause in JJA, associated with elevated PAN upwelling over India from the summertime Asian monsoon. In MAM and SON, peak PAN concentrations range from 120-160 pptv at approximately 10-40°N and 20-30°S, linked to biomass burning over central and southern Africa, respectively. In SON, there is enhanced PAN (70-90 pptv) between 70-90°S in the Antarctic lowermost stratosphere.

The UoL zonal mean PAN concentrations (Figure 6) are smaller in the troposphere and in the vicinity of the dynamical tropopause, although they do have have similar spatial patterns to the IMK data. In JJA, the peak UoL PAN near the northern mid-latitude tropopause, linked to the summer-time Asian monsoon, is between 100 - 170 pptv. Similar biomass burning signals occur in MAM and SON, but again the concentrations of between 90-150 pptv are lower than IMK data. The UoL retrievals also show high PAN concentrations between 200-100 hPa in the SON southern high latitudes, but the magnitude is less pronounced than in the IMK data.



We have compared both MIPAS PAN retrievals to ACE-FTS zonal mean profiles (Figure 7) for 2007-2008. Although there is no validation of the ACE-FTS PAN product independent of the UoL MIPAS data set considered here, we use it for further assessment of both MIPAS PAN products to try and evaluate the differences between them. In 2007-2008 there were approximately 5000 ACE-FTS PAN retrievals, which we co-located with corresponding MIPAS retrievals. For this comparison each MIPAS retrieval had to be within 6 hours and 1000 km of the ACE-FTS retrievals.

In the tropical regions (30°S - 30°N), the UoL MIPAS PAN and ACE-FTS PAN concentrations are similar between 70-90 pptv at 200 hPa; IMK MIPAS tends to larger (>100 pptv). However, the IMK MIPAS and ACE-FTS PAN profiles converge in the LS and the UoL MIPAS PAN is lower by 20-30 pptv. At 30-60°N and S, the IMK MIPAS PAN is higher than the other products by 20-40 pptv between 200-175 hPa. At 150-100 hPa, all three vertical profiles range between 30-50 pptv. Above 100 hPa, ACE-FTS PAN overestimates MIPAS PAN by about 20-30 pptv as the two MIPAS profiles converge. However, MIPAS sensitivity is reduced at these altitudes and PAN retrievals are heavily dependent on the apriori. Finally, between 60-90°N and S, where the concentrations are generally the lowest globally, there is little difference in the MIPAS profiles at 200 hPa (ACE-FTS PAN is lower by 20 pptv at 60-90°S). Between 150-75 hPa, the UoL MIPAS PAN concentrations are larger than the IMK values by 10-40 pptv, with mixed agreement with the ACE-FTS PAN profiles in this altitude range. Above 75 hPa, there are large differences (50 pptv) between the IMK and UoL MIPAS PAN in the southern hemisphere. Here, the IMK MIPAS and ACE-FTS PAN profiles are in better agreement. In the northern hemisphere, both MIPAS products are in better agreement as the ACE-FTS PAN profile is 10-30 pptv higher. Overall, despite the differences in the satellite PAN retrievals, all three products largely fall within the uncertainty ranges of each other.



## 3.2 IMK - UoL Differences

Reasons for the differences between the IMK and UoL MIPAS PAN retrievals are potentially linked with the independent retrieval schemes. The UoL MIPAS Orbital Retrieval using Sequential Estimation (MORSE) scheme is an optimal estimation algorithm in logarithmic parameter space with MOZART PAN values as constraints for the profile regularisation. The IMK retrieval uses a $1^{st}$ order Tikhonov regularisation which constrains the differences between adjacent profile values towards small values, i.e. the constraint does not directly influence the profile values but rather the smoothness of the retrieved profile. Furthermore the two schemes use different forward models to calculate the radiative transfer. The IMK retrieval utilises the Karlsruhe Optimized and Precise Radiative transfer Algorithm (KOPRA), while the MORSE scheme uses a version of the Reference Forward Model (RFM). A previous study (Glatthor et al., 1999) found that differences in the KOPRA and RFM interpolation approach for cross-section data gave differences in CFC-12 results of up to 30 nW/(cm$^2$ sr cm$^{-1}$), which is comparable with the MIPAS noise equivalent spectral radiance (NESR) in band A. PAN data are in the form of cross-sections, although no equivalent test has been carried out for this species to test the expected radiance difference.

Alongside the forward models used there are also several differences in the retrieval set-up which may account for some of the differences. The optimised resolution MIPAS data are measured on levels which are approximately 1.5 km apart in the UTLS. The MORSE state vector retrieves on the same 1.5 km spaced levels, whereas the IMK retrieval is on a finer 1 km grid. The IMK retrieval also uses one single retrieval microwindow (775-800 cm$^{-1}$, but is split into two sub-microwindows of 775-787 cm$^{-1}$ and 794.5-800 cm$^{-1}$), whereas the MORSE retrieval uses 5 smaller windows in the 777 cm$^{-1}$ to 798 cm$^{-1}$ range which are ordered in terms of simulated information content to use the window with highest information content for the first fit. These are slightly different to the windows used in the full-resolution mode in Moore and Remedios (2010) and are (1) 784.9375 cm$^{-1}$ to 787 cm$^{-1}$, (2) 779.5 cm$^{-1}$ to 784.125 cm$^{-1}$, (3) 777.25



cm$^{-1}$ to 779.125 cm$^{-1}$, (4) 794 cm$^{-1}$ to 795.75 cm$^{-1}$ and (5) 796.0625 cm$^{-1}$ to 797.75 cm$^{-1}$. Both schemes fit continua in the retrieval process and fit offsets to each retrieval microwindow.

Interfering species are handled differently: MORSE performs sequential retrievals, meaning that each species is retrieved in turn. For the MORSE PAN, the order is p, T, $H_2O$, $O_3$, $HNO_3$, $ClONO_2$, and $CCl_4$ before retrieval of PAN. The IMK processor also performs sequential retrievals, but from these only the pre-fitted species p, T, $HNO_3$, ClO, F-11, $C_2H_6$, and HCN are used in the PAN retrieval, while $CH_3CCl_3$, $CCl_4$, $ClONO_2$, F-22, $O_3$, $H_2O$, and $C_2H_2$ are fitted together with PAN in the same microwindow.

## 3.3 Model - Aircraft Comparisons

Figure 8 shows the comparisons between TOMCAT and the aircraft measurements from the ARCPAC and ARCTAS campaigns for CO, PAN, acetone and acetaldehyde in 2008. TOMCAT output has been interpolated both spatially and temporally to the location and time of the observations. The observed and modelled median concentration in 50-hPa pressure bins are used to give a vertical profile. The $25^{th}$ and $75^{th}$ percentiles for both the model and observations are shown to indicate the spread of the model and observations within each bin. For the ARCTAS data results from two different measurements of acetone and acetaldehyde by different techniques (Proton Transfer Reaction Mass Spectrometry, PTRMS, and Trace Organic Gas Analyzer, TOGA) are shown.

When compared with the ARCPAC campaign, TOMCAT springtime CO is low throughout the troposphere which is a common problem in global models at higher latitudes (Monks et al., 2015). However, there is also evidence of a plume of enhanced CO that is not captured by the model at 600 hPa. PAN is also clearly enhanced at about 600 hPa, which again is not captured by the model. During April 2008 there were unusually high emissions from biomass burning that were transported to the Arctic. The ARCPAC campaign, targeted some of these plumes leading to enhanced measurements of several species (Warneke et al., 2010). The inability of the model to capture these



enhancements is likely due to the biomass burning emissions used in the model or its coarse horizontal resolution and it is difficult to draw any conclusions about TOMCAT PAN here.

In the ARCTAS summer campaigns (ARCTAS-B & ARCTAS-CARB), TOMCAT successfully reproduces the aircraft CO profile. For PAN, the TOMCAT average profile is within the ARCTAS variability range apart from at 950 hPa (+300 pptv, ARCTAS-B) and 800-750 hPa (-100 pptv, ARCTAS-CARB), but captures UT PAN successfully. Compared with ARCTAS-A, TOMCAT significantly overestimates PAN by 150-200 pptv between 950-700 hPa and by 20-50 pptv at 450-250 hPa. Between 700-450 hPa and above 250 hPa, TOMCAT PAN is within the observational variability. TOMCAT acetaldehyde average profiles underestimate the ARCTAS-A, B and CARB profiles in the mid-lower troposphere. Emmons et al. (2015) found that several models underestimated acetaldehyde from this campaign in spring (including TOMCAT), but in summer TOMCAT concentrations were on the low end of the model distribution. Acetone was also found to be low in these models in summer when compared with this data. However, in spring there was a wide range in acetone in the same models suggesting the springtime low bias in acetone is a problem in TOMCAT. The models which had higher acetone also had lower PAN suggesting TOMCAT may be too efficient at producing PAN during long-range transport events to the Arctic. If acetone sources were increased in the model this would likely make PAN concentrations too high.

We also compare TOMCAT with the multi-year regional aircraft composite dataset compiled by Emmons et al. (2000), which allows for comparisons in other regions. Within this dataset, aircraft profiles for several geographic regions are constructed using data from several flights representing large spatial and temporal averages. TOMCAT output for 2007-8 was averaged over the same spatial regions and months as each of the aircraft profiles. Given the climatological nature of the aircraft profiles, and the high degree of variability exhibited by tropospheric PAN, the aircraft profiles may not be truly representative of the distribution in a given region for the simulated period used in this study. With this in mind, profiles were selected for comparison which are



likely to be representative of background concentrations in a particular region. A disadvantage with this method is the temporal difference between the TOMCAT runs and the Emmons et al. (2000) climatology.

In Figure 9, TOMCAT reproduces the vertical structure of aircraft PAN in Hawaii, but significantly overestimates PAN throughout the profile at Alaska. At Christmas Island, TOMCAT and aircraft data agree well in the lower - mid troposphere, but the model significantly overestimates above 5 km. Near the surface, TOMCAT is able to reproduce the low PAN concentrations where there are no sources. In the more anthropogenically polluted regions, e.g. Japan and China, the model struggles to simulate the larger near-surface PAN concentrations. In the boundary layer, model PAN increases with altitude, while aircraft profiles decrease. However, TOMCAT PAN is within the observational variability and captures the vertical structure of PAN above 2 km. Near the US East coast, TOMCAT captures the near surface concentrations (approximately 1000 pptv), but overestimates PAN in the lower-mid troposphere by 200-500 pptv. In the regions of strong biomass burning signals, TOMCAT captures the vertical structure within the aircraft uncertainty range at the West African coast but significantly underestimates PAN in East Brazil.

Overall, the above figures show that TOMCAT can generally reproduce UT PAN observed from the ARCTAS campaign in the spring and summer of 2008, although these comparisons are limited to North America. Comparisons with the Emmons et al. (2000) climatology show that TOMCAT can capture the majority of the PAN vertical profiles in various global background regions. Therefore, we have confidence in the model and use it as a tool to assess differences in the IMK and UoL MIPAS PAN products.

## 3.4 TOMCAT - Satellite Comparisons

TOMCAT, with the MIPAS AKs applied, at 150 hPa (Figures 10 and 11) has maximum PAN concentrations in the UT ($>$100 pptv) over the tropics and minimum values ($<$100 pptv) in the LS over the mid-high latitudes. In DJF (Figures 10 and 11), TOMCAT has elevated PAN (130-150 pptv) over central Africa like MIPAS, but the largest model PAN





values are over tropical South America and South East Asia (150-180 pptv). Such features are not as noticeable in the MIPAS data sets. In MAM, TOMCAT reproduces the biomass burning PAN signal (120-150 pptv) over central Africa, although this is lower than IMK and UoL PAN values in this region. In JJA, TOMCAT has elevated PAN

5 concentrations over India linked to convective upwelling of PAN into the UTLS from the summer-time Asian monsoon. This signal is clearly seen in the IMK MIPAS PAN data, but less so in the UoL data. The peak TOMCAT PAN concentrations (170-200 pptv) are over the Middle East, which is also seen by the MIPAS PAN datasets. In SON, TOMCAT misses PAN associated with biomass burning plumes from southern

10 Africa, which propagate out into the South Atlantic. In the IMK and UoL PAN products, PAN concentrations range from 150-200 pptv, while they are only 100-120 pptv in the TOMCAT PAN distribution. Note that the TOMCAT PAN distributions at 150 hPa in Figures 10 and 11 are slightly different due to the application of the IMK and UoL MIPAS AKs. Typically, with the application of the IMK MIPAS PAN AKs, the TOMCAT PAN

15 concentrations are larger in the UT and lower in the LS when compared to TOMCAT PAN concentrations with the UoL AKs applied.

Figures 12 and 13 show the differences between the satellite observations and TOMCAT simulations for the IMK and UoL retrievals, respectively. Again the purple polygonned regions show where the differences are significant, i.e. where the absolute

20 model-satellite mean bias (MB) is greater than that of the observational error. In DJF, TOMCAT significantly overestimates IMK PAN by 30-60 pptv throughout the tropical UT region, apart from Africa. Though the largest differences are over tropical South America and South East Asia. There are significant negative biases of -20 to 0 pptv in the LS, which occur in all seasons. In MAM, the largest differences of -90 to -60 pptv

25 are over central Africa. Though TOMCAT captures the biomass burning signal in MAM (Figure 10), it still significantly under predicts the IMK MIPAS PAN. In JJA and SON, TOMCAT significantly underestimates ($<$-50 pptv) IMK MIPAS PAN across the majority of the domain, especially in the northern mid-latitudes and southern Africa and the South Atlantic, respectively.



When compared with the UoL MIPAS data (Figure 13), TOMCAT generally under-estimates PAN in the LS, while overestimating it in the UT. In MAM, JJA and SON, TOMCAT is significantly biased by -40 to 0 pptv in the LS. In DJF, this signal is reduced in the northern high latitudes and is positive in the southern high latitudes. The largest TOMCAT PAN underestimation of between -80 to -50 pptv is in SON over southern Africa. Here, TOMCAT seems to be missing PAN produced from $NO_x$ biomass burning emissions, which is seen the in IMK data. The large positive biases in DJF (30-70 pptv), also seen in Figure 12, are over South East Asia, the Pacific and Central/South America. In MAM, significant positive biases are typically between the Equator and the southern dynamical tropopause.

Zonal mean PAN TOMCAT, with both sets of AKs applied, is shown in Figures 14 and 15. In all seasons, PAN ranges between 0-50 pptv in LS and 50-100 pptv around the tropopause. In the UT, TOMCAT PAN ranges between 100-150 pptv in DJF, MAM and SON. In JJA, peak PAN concentrations are larger and reach 160-180 pptv, linked to the summertime Asian monsoon. When compared with MIPAS zonal PAN, TOMCAT does not have the same elevated concentrations associated to the MAM and SON African biomass burning signals. Similar to the 150 hPa comparisons (Figures 10 and 11), TOMCAT PAN concentrations with the IMK MIPAS PAN AKs applied are higher in the UT and lower in the LS than the UoL equivalent.

The TOMCAT - satellite differences in zonal mean PAN are shown in Figures 16 and 17. Here, the hatching shows regions of non-significant differences. As for the differ-ences at 150 hPa, TOMCAT significantly underestimates IMK PAN by 10 to >80 pptv in the LS. In DJF, TOMCAT simulates higher PAN concentrations (0-50 pptv) than ob-served between 200-125 hPa at 10°S-30°N. Negative biases (-50 to -40 pptv) around the NH and SH dynamical tropopause in MAM and SON, respectively, are linked to lower TOMCAT PAN concentrations in regions of biomass burning. In JJA, TOMCAT underestimates IMK MIPAS PAN throughout the mid-latitudes between 200-100 hPa, as seen in Figure 12. When compared with UoL MIPAS PAN, TOMCAT significantly overestimates PAN in the UT by 20-60 pptv in DJF between 200-100 hPa. In JJA and





SON, significant positive biases (20-40 pptv) occur near the NH tropopause. In the LS, TOMCAT significantly underestimates MIPAS by 0-30 pptv in most seasons.

Overall, TOMCAT significantly underestimates IMK and UoL MIPAS PAN in the LS in all seasons (except for UoL MIPAS PAN in DJF). In the UT, TOMCAT tends to significantly underestimate IMK MIPAS PAN, especially in the biomass burning regions. In DJF, the TOMCAT PAN concentrations are large compared with both MIPAS PAN products over tropical South America and South East Asia. When compared with UoL MIPAS PAN in the UT, TOMCAT overestimates by 10-90 pptv. Typically, there is some consistency between the two MIPAS products in the LS. However, in the UT, the IMK MIPAS PAN concentrations are larger than the UoL with TOMCAT values in between them. Fadnavis et al. (2014) found that ECHAM5-HAMMOZ simulations underestimated IMK MIPAS PAN concentrations in the summer-time Asian monsoon. Emmons et al. (2015) and Arnold et al. (2015) found that TOMCAT overestimates aircraft observed PAN in the troposphere. Emmons et al. (2015) found biases between -10% to +30% between 3-7km, with a large bias occurring against some springtime flights (+80%). As shown here, the largest TOMCAT biases are in spring (Fig 8), but differences are generally within the variability of the aircraft observations. Even though these comparisons are not at altitudes observed by satellite, it quantifies the skill of TOMCAT and allows us to use the model as a tool to better understand UTLS PAN. This gives us confidence to state that there are inconsistencies between the two MIPAS PAN datasets as IMK and UoL MIPAS PAN are positively and negatively biased with the model in the UT.

## 4  Conclusions

We have compared two independent MIPAS retrievals of PAN which are produced by IMK, Karlsruhe and the University of Leicester. We analysed observations for the 2-year period 2007-2008 in the upper troposphere and lower stratosphere. Overall, the IMK MIPAS PAN has significantly larger concentrations in the upper troposphere over



the tropics by 50-100 pptv, when compared with UoL data. In the lower stratosphere, the UoL concentrations are larger by 0-30 pptv, however, these differences are only significant in the northern high latitudes in MAM and JJA. Both retrieved datasets show peak PAN concentrations over the African biomass burning regions ($>$200 pptv), but the IMK data has a clearer summertime Asian monsoon signal. Here, enhanced convection leads to increased vertical transport of PAN into the UTLS and the outflow ranges from 150 to $>$200 pptv. When compared with PAN from ACE-FTS, the MIPAS profile uncertainties generally overlap with those from the ACE-FTS in the UTLS. At 200-175 hPa, IMK MIPAS PAN tends to overestimate the other two products. Between 75-25 hPa, the ACE-FTS PAN concentrations tend to be larger than the MIPAS profiles (though in agreement with IMK MIPAS PAN at 30-60°N and S).

The TOMCAT global CTM was used to help quantify the global distribution of PAN. At 150 hPa, TOMCAT significantly underestimates upper tropospheric IMK MIPAS PAN by 50 to $>$100 pptv in the biomass burning regions in MAM and SON. It also underestimates the observed lower stratospheric PAN in all seasons. When compared with UoL MIPAS PAN, TOMCAT significantly overestimates the observations by 10-70 pptv in the upper troposphere (tropics) and underestimates by 10-40 pptv in the lower stratosphere (mid-high latitudes). Previous publications (e.g. Emmons et al. (2015)) have shown that TOMCAT overestimates PAN in the troposphere and the comparisons between TOMCAT and aircraft data in this study show similar patterns in the spring ARCTAS campaign, when lower tropospheric PAN is particularly stable and long-lived, and at several regions in the Emmons et al. (2000) aircraft climatology. However, the model does a good job at capturing PAN during summer. In the UTLS, TOMCAT PAN reproduces the observations, given the large uncertainty in aircraft measurements.

Based on the inter-comparison of satellite products and comparison of TOMCAT with observations, we suggest that there are inconsistencies between the two MIPAS PAN datasets as IMK and UoL MIPAS PAN are positively and negatively biased with the model in the upper troposphere.

*Acknowledgements.* This work was supported by the NERC National Centre for Earth Obser-



vation (NCEO). We are grateful to Paul Young (University of Lancaster) for supplying the TOM-CAT isoprene scheme. We acknowledge the use of Emmons et al. (2000) aircraft climatology of atmospheric trace gases, which is available from https://www2.acom.ucar.edu/gcm/aircraft-climatology. We also thank the NOAA Earth System Research Laboratory - Chemical Sciences Division for the ARCPAC CO aircraft data. The ACE mission is funded primarily by the Canadian Space Agency (http://www.asc-csa.gc.ca/eng/). PTR-MS measurements during ARCTAS were funded through the Austrian Space Applications Programme (ASAP). ASAP is sponsored by the Austrian Ministry for Transport and administered by the Aeronautics and Space Agency (ALR) of the Austrian Research Promotion Agency (FFG). Tomas Mikoviny is acknowledged for his support in the PTR-MS data acquisition and analysis.

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





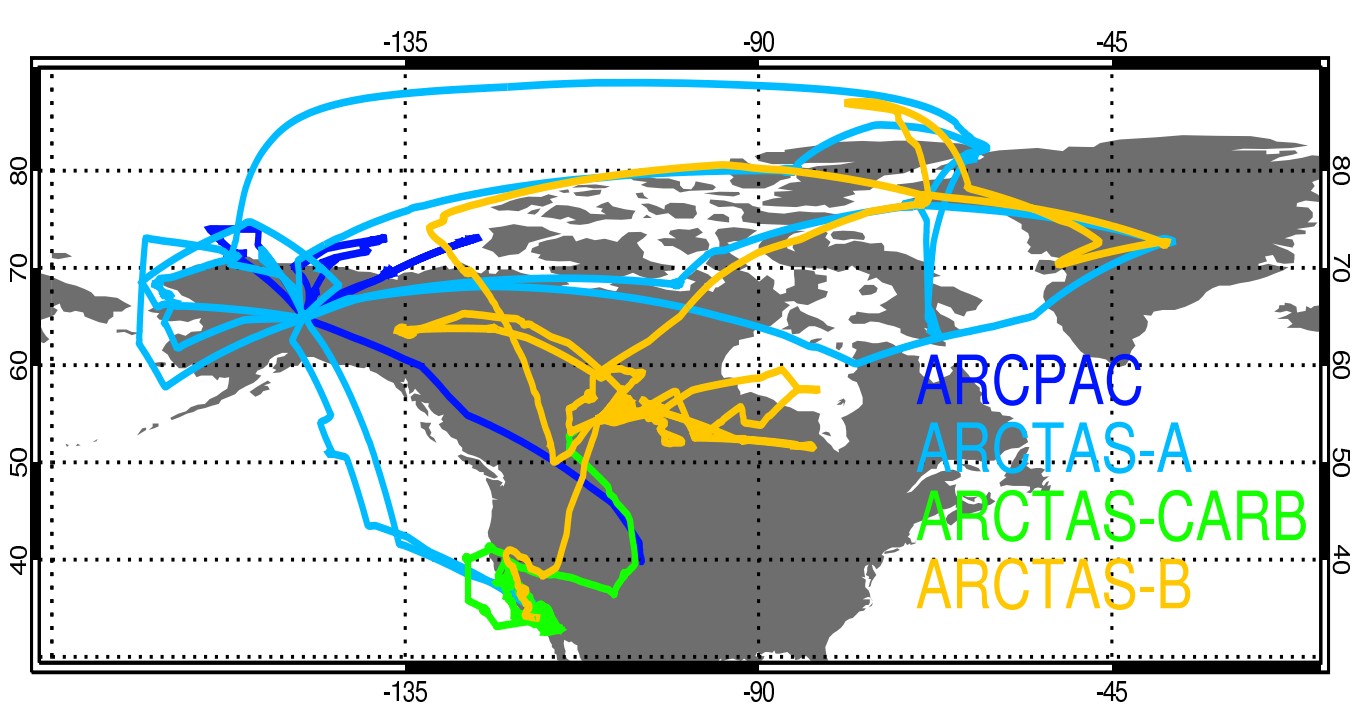

**Fig. 1.** Map of the flightpaths of the ARCPAC and ARCTAS aircraft campaigns in 2008 used to evaluate TOMCAT PAN and precursor species. See text for more details.





**Fig. 2.** MIPAS PAN (pptv) from the IMK retrieval at 150 hPa for 2007-2008 in a) December-January-February (DJF), b) March-April-May (MAM), c) June-July-August (JJA) and d) September-October-November (SON). The black dashed lines show the dynamical tropopause (defined as ±2 PVU) based on ERA-Interim data.







**Fig. 3.** As Figure 2 but for the UoL MIPAS retrieval.



**Fig. 4.** Difference in MIPAS PAN (pptv) from IMK retrieval minus UoL retrieval at 150 hPa for 2007-2008 in a) DJF, b) MAM, c) JJA and d) SON. Purple polygonned regions show regions of significant differences, where the IMK and UoL retrieval uncertainty ranges do not overlap.



**Fig. 5.** Zonal mean MIPAS PAN (pptv) from the IMK retrieval for 2007-2008 in a) DJF, b) MAM, c) JJA and d) SON. The black dashed lines show the dynamical tropopause (defined as ±2 PVU) based on ERA-Interim data.





**Fig. 6.** As Figure 5 but for the UoL MIPAS retrieval.



**Fig. 7.** Zonal mean profiles of PAN (pptv) averaged within different latitude bands for 2007-2008 from ACE-FTS (green line), IMK MIPAS (red) and UoL MIPAS (blue). Horizontal lines give the satellite uncertainty ranges. MIPAS retrievals have been co-located with ACE-FTS retrievals.



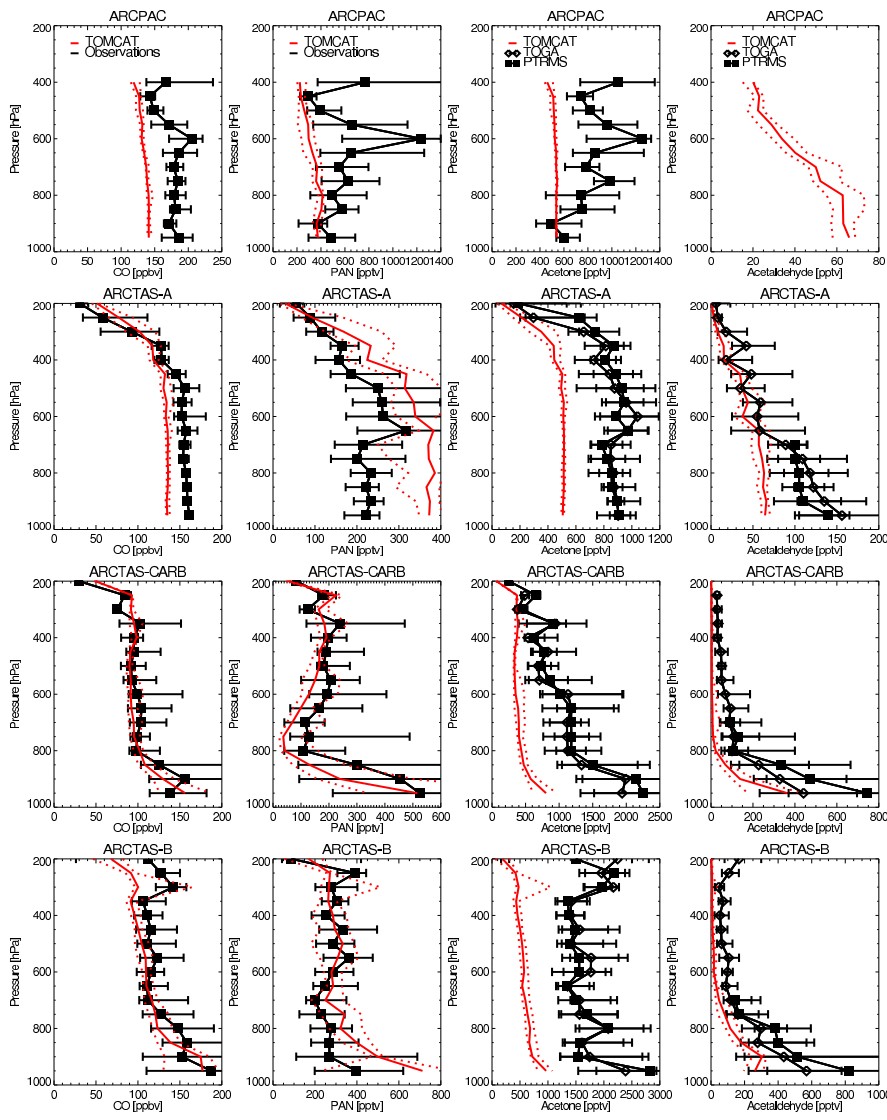

**Fig. 8.** Observed profiles of CO (ppbv), PAN, acetone and acetaldehyde (pptv) for 2008 from the ARCPAC and ARCTAS campaigns compared to results from the TOMCAT model sampled in the same location. The black lines give the median observed concentration and the error bars give the $25^{th}$ & $75^{th}$ percentiles. The solid red line gives the median modelled concentration and the dotted lines give the $25^{th}$ & $75^{th}$ percentiles.




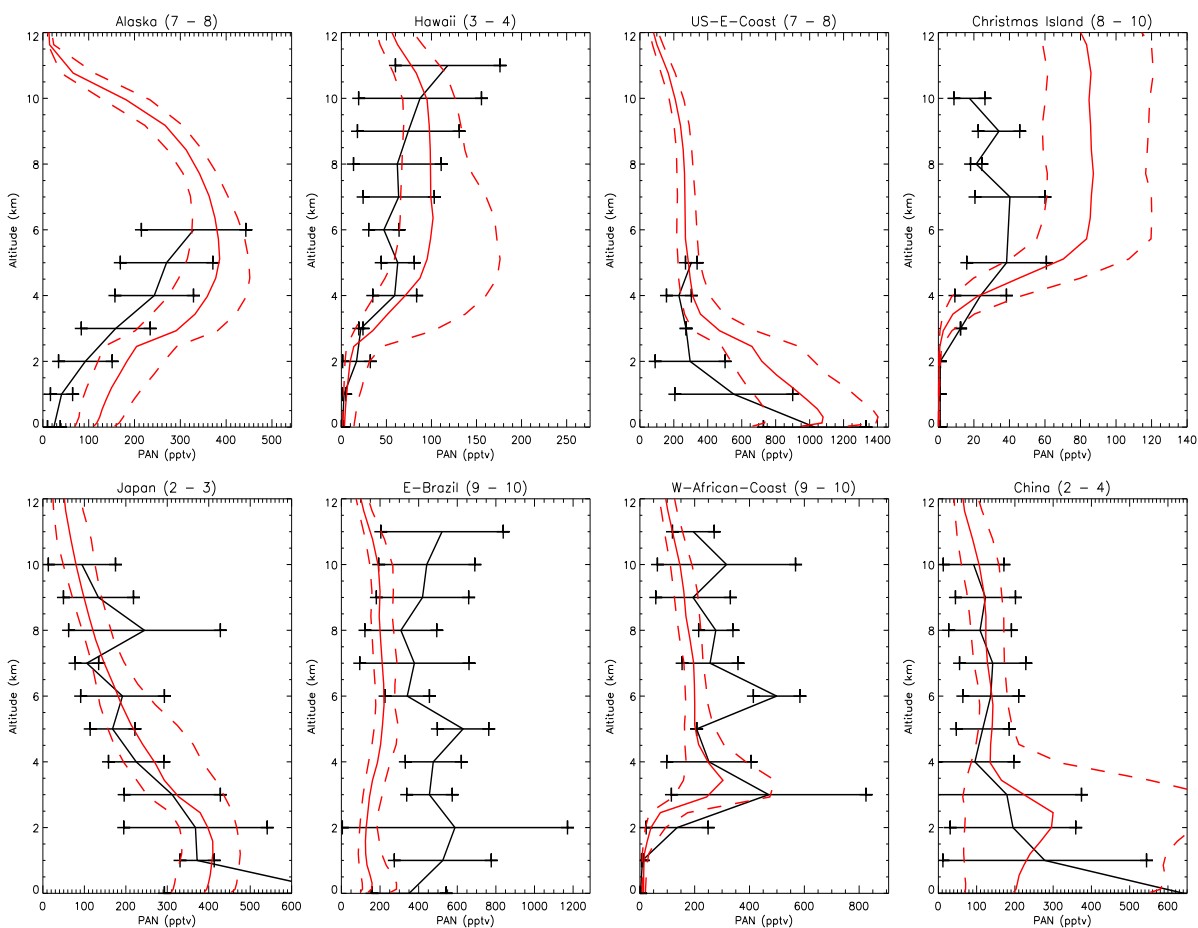

**Fig. 9.** Vertical profiles of PAN from the Emmons et al. (2000) aircraft climatology (black) and the TOMCAT model (red) for eight different regions given in the panel titles The numbers in the title represent the months sampled. The TOMCAT PAN data has been averaged over the same locations and times as the measurements. The horizontal error bars are the observational standard deviations and the dashed lines are the model $25^{th}$ & $75^{th}$ percentiles.





**Fig. 10.** Mixing ratio of PAN (pptv) from the TOMCAT model (with the IMK averaging kernels applied) at 150 hPa averaged over the periods a) DJF, b) MAM, c) JJA and d) SON in 2007-2008.





**Fig. 11.** As Figure 10 but for the application of the UoL averaging kernels to the TOMCAT model output.





**Fig. 12.** Difference in PAN (pptv) between the TOMCAT model (with IMK AKs applied) and observed IMK MIPAS PAN at 150 hPa for 2007-2008 in a) DJF, b) MAM, c) JJA and d) SON. Purple polygonned areas show regions of significant differences, where the |TOMCAT - observations| > observational error.





**Fig. 13.** As Figure 12 but for UoL MIPAS observations and application of the UoL averaging kernels to the TOMCAT model output.



**Fig. 14.** Zonal mean PAN (pptv) from the TOMCAT model (with IMK AKs applied) for 2007-2008 in a) DJF, b) MAM, c) JJA and d) SON.





**Fig. 15.** As Figure 14 but for the application of the UoL averaging kernels to the TOMCAT model output.







**Fig. 16.** Difference in zonal mean PAN (pptv) between the TOMCAT model (with IMK AKs applied) and observed IMK MIPAS PAN for 2007-2008 in a) DJF, b) MAM, c) JJA and d) SON. Hatching represents non-significant differences.



**Fig. 17.** As Figure 16 but for UoL MIPAS observations and application of the UoL averaging kernels to the TOMCAT model output.