# Peer review of "Intercomparison and evaluation of satellite peroxyacetyl nitrate observations in the upper troposphere - lower stratosphere"

_Atmospheric Chemistry and Physics, 2016_

## Referee Comment (RC1) · Anonymous Referee #1 · 11 Jul 2016

Comments on "Intercomparison and evaluation of satellite peroxyacetyl nitrate observations in the upper troposphere - lower stratosphere" by Pope et al.

The paper provides detailed intercomparison of peroxyacetyl nitrate from satellite observations (MIPAS and ACE-FTS), Aircraft and chemical transport model (TOMCAT). This paper provides interesting results on comparison during all the seasons. The paper is well written and can be accepted for publication in ACP after the minor revision. I suggest authors to incorporate following suggestions.

(1)Authors should add a statement as to why two-year time period 2007-2008 is used for inter comparison.

[Figure]

(2) Although authors have cited reference, readers should know the difference between retrieval methods used by the University of Leicester (UoL) and the Institute of Meteorology and Climate Research, Karlsruhe Institute for Technology (IMK). A brief description will be appreciated.

(4) A brief description on 'uncertainty in IMK MIPAS PAN retrieval' should be added.

(5) conclusion sections should provide quantitative statements on comparison.

(6) X-axis and y-axis labels in figures 2, 3, 4,7,8,9, 10,11,12,13 are not clear.
* * *

---

## Referee Comment (RC2) · Anonymous Referee #2 · 25 Jul 2016

Overview: The paper provides a comparison of PAN measurements from the MIPAS instrument on ENVISAT produced using two different retrieval methodologies. It also includes a comparison of the retrievals (and a suite of aircraft profiles) to a TOMCAT simulation of PAN. The paper is technically strong, and well written. Unfortunately, the paper does not really further our understanding of the sources of PAN in the atmosphere in any significant way. This makes it difficult to review, because there are no substantive conclusions drawn. That being said, this paper does represent the first presentation of a new dataset, which could have significant value. Thus, I recommend publication of the paper, but recommend that the authors consider the following idea for improvement. Both datasets (Figure 2 and Figure 3) show a consistent PAN maxi-

mum in the tropical UT in SON. Can the authors use the data to attribute this feature? Casual mention is made throughout the text that the source is biomass burning, but no evidence is provided that this is the source. What does the model attribute this feature to? Is there a way to use any other simultaneously retrieved tracers (i.e. HCN or CO) to better understand whether the presence of this feature is driven by biomass burning or by lightning NOx forming PAN in the presence of isoprene oxidation products lifted in convection?

I have only one minor comment. On Page 4, Line 21: PAN mixing ratios are on the order of several ppbv in heavily polluted air-masses. This sentence is strange here.

---

## Author Response (AR1)

**Response to Reviewer 1's Comments:**

*We thank the reviewer for their comments. The reviewer's comments are in black text and our responses are in red text. Any additions to the manuscript are in blue text. Reference to page and line numbers refer to the original manuscript.*

Comments on "Intercomparison and evaluation of satellite peroxyacetyl nitrate observations in the upper troposphere - lower stratosphere" by Pope et al.

The paper provides detailed intercomparison of peroxyacetyl nitrate from satellite observations (MIPAS and ACE-FTS), Aircraft and chemical transport model (TOMCAT). This paper provides interesting results on comparison during all the seasons. The paper is well written and can be accepted for publication in ACP after the minor revision. I suggest authors to incorporate following suggestions.

(1) Authors should add a statement as to why two-year time period 2007-2008 is used for inter-comparison.

We use the years 2007-2008 because this is the time period where we had available data for from multiple datasets (i.e. MIPAS, ACE-FTS and the aircraft data). We have added the following text on page 9 line 6, "We perform TOMCAT simulations for 2007-2008, since MIPAS, ACE-FTS and aircraft data are available for this period."

(2) Although authors have cited reference, readers should know the difference between retrieval methods used by the University of Leicester (UoL) and the Institute of Meteorology and Climate Research, Karlsruhe Institute for Technology (IMK). A brief description will be appreciated.

On page 6, lines 17-19, we have altered "The UoL and IMK MIPAS PAN retrieval methods are discussed by Moore and Remedios (2010) and Glatthor et al. (2007), respectively." to "The UoL MIPAS PAN retrieval is based on an optimal estimation scheme in logarithmic parameter space, while the IMK MIPAS PAN retrieval consists of inversion of level-1B spectra to vertical profiles of atmospheric state parameters by constrained non-linear least squares fitting in a global-fit approach. The constraint is implemented as a $1^{st}$ order Tikhonov regularisation with an all-zero a-priori profile. The two MIPAS retrieval schemes are discussed in more detail by Moore and Remedios (2010) and Glatthor et al. (2007), respectively, and compared in section 3.2.".

(4) A brief description on 'uncertainty in IMK MIPAS PAN retrieval' should be added.

The IMK and UoL MIPAS PAN errors are discussed on page 10 and 11, lines 18-29 and 1-3 respectively. Differences in the retrievals which might be causing the product biases are discussed in section 3.2. Therefore, the IMK and UoL MIPAS errors are discussed to an equal extent. We have added some information on page 10, line 22 to outline the sources of errors in the retrievals. "Sources of retrieval error include measurement noise, interfering signals from other trace gases, errors in the temperature profile, instrument pointing, spectroscopic errors, calibration errors and instrumental line of shape (Glatthor et al., 2007).".

(5) conclusion sections should provide quantitative statements on comparison.

We feel that the conclusions have suitable levels of quantitative information in them. Throughout the conclusions, we quantitatively highlight the important differences and similarities between datasets. More information could be added about the results from previous sections, but this would defeat the object of the conclusions where concise statements are required.

Where the MIPAS - ACE-FTS and TOMCAT – aircraft comparisons are summarised, these are important, but secondary results in the paper and discussed is less detail. Again, to try and keep the conclusions succinct.

(6) X-axis and y-axis labels in figures 2, 3, 4,7,8,9, 10,11,12,13 are not clear.

We have increased the axis label sizes in line with the reviewers comment.

**References:**

Glatthor, N., von Clarmann, T., Fischer, H., Funke, B., Grabowski, U., Höpfner, M., Kellmann, S., Kiefer, M., Linden, A., Milz, M., Steck, T., and Stiller, G. P.: Global peroxyacetyl nitrate (PAN) retrieval in the upper troposphere from limb emission spectra of the Michelson Interferometer for Passive Atmospheric Sounding (MIPAS), Atmospheric Chemistry and Physics, 7, 2775–2787, doi:10.5194/acp-7-2775-2007, 2007.

**Response to Reviewer 2's Comments:**

*We thank the reviewer for their comments. The reviewer's comments are in black text and our responses are in red text. Any additions to the manuscript are in blue text. Reference to page and line numbers refer to the original manuscript.*

Overview: The paper provides a comparison of PAN measurements from the MIPAS instrument on ENVISAT produced using two different retrieval methodologies. It also includes a comparison of the retrievals (and a suite of aircraft profiles) to a TOMCAT simulation of PAN. The paper is technically strong, and well written. Unfortunately, the paper does not really further our understanding of the sources of PAN in the atmosphere in any significant way. This makes it difficult to review, because there are no substantive conclusions drawn. That being said, this paper does represent the first presentation of a new dataset, which could have significant value. Thus, I recommend publication of the paper, but recommend that the authors consider the following idea for improvement.

We are happy to read that the reviewer feels the manuscript is suitable for publication in ACP. We would argue though that there are substantial scientific conclusions in our paper. Both the IMK and UoL groups have published papers on their MIPAS PAN retrievals and other studies have used this data (e.g. Fadnavas et al., 2014) or cited it (e.g. Fischer et al., (2014)). Therefore, our study highlights that there are inconsistences (page 20, lines 25-28) between these datasets and this needs to be acknowledged by future studies when discussing PAN concentrations in the UTLS retrieved from satellite. We also provide potential reasons for this in section 3.2. And finally, as the reviewer points out, we do include a new dataset (i.e. global investigation of PAN retrieved from ACE-FTS) in the manuscript.

Both datasets (Figure 2 and Figure 3) show a consistent PAN maximum in the tropical UT in SON. Can the authors use the data to attribute this feature? Casual mention is made throughout the text that the source is biomass burning, but no evidence is provided that this is the source. What does the model attribute this feature to? Is there a way to use any other simultaneously retrieved tracers (i.e. HCN or CO) to better understand whether the presence of this feature is driven by biomass burning or by lightning NOx forming PAN in the presence of isoprene oxidation products lifted in convection?

This is a good point as both mechanisms will result in PAN formation. Unfortunately, as shown in Figures 12c and 13c, the model struggles to capture the enhanced PAN signal over the tropical UT in SON with significant negative biases in both cases. Therefore, it is difficult to use the model to try and diagnose the potential source of the enhanced PAN in this region.

[Figure]

**Figure SI**: Atmospheric composition over the tropical South Atlantic observed by satellite in Sept-Oct-Nov 2008. a) Ozone Monitoring Instrument (OMI; Boersma et al., 2011) tropospheric column $NO_2$ ($10^{15}$ molecules/cm$^2$), b) OMI-MLS (Microwave Limb Sounder; Ziemke et al., 2006) tropospheric $O_3$ (DU), c) Michelson Interferometer for Passive Atmospheric Sounding (MIPAS) HCN (pptv) at 150 hPa and d) MIPAS PAN (pptv) at 150 hPa (Glatthor et., 2007). White circles represent locations of significant lightning events detected by the Lightning Imaging Sensor (LIS; Cecil et al., 2014).

In Figure SI, we have plotted SON 2008 averages of OMI tropospheric $NO_2$, tropospheric OMI/MLS $O_3$ and MIPAS HCN and PAN at 150 hPa. From Figure SIa), there are clear source regions of $NO_2$ over both southern Africa and South America, with transport of $NO_2$ out into the tropical South Atlantic. Tropospheric $O_3$ (Figure SIc) has a similar and more distinct pattern over the Atlantic. As enhanced MIPAS PAN (Figure SId) is also seen in this region, $NO_2$, $O_3$ and PAN are highly correlated. The gradient in retrieved tropospheric $O_3$ across the tropical Atlantic, suggests transport of PAN and $NO_2$ and subsequent $O_3$ formation, primarily from southern Africa (i.e. the ozone gradient is from East to West across the Atlantic; Figure SIc). Figure SIb shows MIPAS HCN, with peak concentrations over the southern Atlantic. HCN is produced by biomass burning with a tropospheric lifetime of approximately 5 months (Li et al., 2009). The HCN distribution suggests African biomass burning sources play a key role in the UTLS PAN budget in this region. White circles in Figure SId show locations of significant lightning events (> 5 x $10^6$ W/ster/m$^2$/µm). These are clustered mainly over Southern African and South America. These are also potential sources of $NO_2$ in the mid-upper troposphere in this region. Belmonte Rivas et al., (2015) show that $NO_2$ sub-columns in regions of lightning activity, using a cloud-slicing technique, range between 0-0.5 x$10^{15}$ molecules/cm$^2$, which can make up a significant proportion of the tropospheric column over the South Atlantic/West African coastline (e.g. 0.5-2.0 x$10^{15}$ molecules/cm$^2$). Fischer et al., (2014), through a modelling study, suggest that lightning $NO_x$ emissions can lead up to 50-60% of PAN formation in October total

column. Therefore, we suggest that both pathways (biomass burning and lightning) lead to the enhanced PAN over the tropical Southern Atlantic.

In order to incorporate some of this discussion into our analysis, we have modified the following text in the paper:

Pages 9-10, Lines 20-22, 1 – "During SON, large PAN concentrations over the South Atlantic (>170 pptv) and along the SH dynamical tropopause (approximately 100 pptv) are linked to outflow from the African biomass burning plume" is replaced with "During SON, large PAN concentrations over the South Atlantic (>170 pptv) and along the SH dynamical tropopause (approximately 100 pptv) are linked to outflow from the African biomass burning plume and from lightning-generated $NO_2$ in the mid/upper troposphere. As shown by Belmonte Rivas et al., (2015), using a cloud slicing technique, there are significantly large sub-columns of $NO_2$ in the mid-upper troposphere co-located with lightning activity. In addition, deep convection transports African biomass burning emissions efficiently to the UT in this region. Fischer et al., (2014) indicate that up to 50-60% of PAN formation in the total column can be attributed to lightning $NO_2$ emissions in their modelling study. IMK MIPAS retrievals of HCN (see Supporting Information; SI), which is a long-lived tracer (5 months; Li et al., 2009) sourced from biomass burning, also shows a strong correlation with PAN in this region. Therefore, it appears that both lightning $NO_x$ and biomass burning act as sources of PAN in this region. This is discussed further in the SI.".

We have added Figure SI as Supporting Information with the following text "From Figure SIa, there are clear source regions of $NO_2$ over both southern Africa and South America, with transport of $NO_2$ out into the tropical South Atlantic. Tropospheric $O_3$ (Figure SIc) has a similar and more distinct pattern over the Atlantic. As enhanced MIPAS PAN (Figure SId) is also seen in this region, $NO_2$, $O_3$ and PAN are highly correlated. The gradient in retrieved tropospheric $O_3$ across the tropical Atlantic, suggests transport of PAN and $NO_2$ and subsequent $O_3$ formation, primarily from southern Africa (i.e. the ozone gradient is from East to West across the Atlantic; Figure SIc). Figure SIb shows MIPAS HCN, with peak concentrations over the southern Atlantic. HCN is produced by biomass burning with a tropospheric lifetime of approximately 5 months (Li et al., 2009). The HCN distribution suggests African biomass burning sources play a key role in the UTLS PAN budget in this region. White circles in Figure SId show locations of significant lightning events (> 5 x $10^6$ W/ster/m$^2$/μm). These are clustered mainly over southern African and South America. These are also potential sources of $NO_2$ in the mid-upper troposphere in this region. Belmonte Rivas et al., (2015) show that $NO_2$ sub-columns in regions of lightning activity, using a cloud-slicing technique, range between 0-0.5 x$10^{15}$ molecules/cm$^2$, which can make up a significant proportion of the tropospheric column over the South Atlantic/West African coastline (e.g. 0.5-2.0 x$10^{15}$ molecules/cm$^2$). Fischer et al., (2014), through a modelling study, suggest that lightning $NO_x$ emissions can lead up to 50-60% of PAN formation in October in the total column. Therefore, we suggest that both pathways (biomass burning and lightning) probably lead to the enhanced PAN over the tropical Southern Atlantic.".

Page 10, Line 13 – "apart from the African biomass burning signals of approximately" with "apart from the African biomass burning/lightning $NO_x$ signals of approximately".

Page 11, Line 19 – "linked to biomass burning over central and southern Africa" with "linked to biomass burning/lightning $NO_x$ over central and southern Africa".

Page 11, Line 25 – "Similar biomass burning signals occur in MAM and SON" with "Similar biomass burning/lightning $NO_x$ signals occur in MAM and SON".

Page 17, Lines 8-12 – "In SON, TOMCAT misses PAN associated with biomass burning plumes from southern Africa, which propagate out into the South Atlantic. In the IMK and UoL PAN products, PAN concentrations range from 150-200 pptv, while they are only 100-120 pptv in the TOMCAT PAN distribution." with "In SON, TOMCAT misses PAN over the South Atlantic, which is likely associated with biomass burning outflow and lightning $NO_x$ from southern Africa. In the IMK and UoL PAN products, PAN concentrations range from 150-200 pptv, while they are only 100-120 pptv in the TOMCAT PAN distribution. This low model bias means that it is difficult to use the model to diagnose the relative contributions of biomass burning and lightning $NO_x$ to the formation of PAN in this region and season.".

Page 18, Lines 6-7 – "produced from $NO_x$ biomass burning emissions, which is seen the in IMK data " with "produced from $NO_x$ biomass burning and lightning emissions, which is seen in the IMK data". The typo has also been corrected.

Page 18, Line 25 – "lower TOMCAT PAN concentrations in regions of biomass burning" with "lower TOMCAT PAN concentrations in regions of biomass burning and peak lightning activity".

I have only one minor comment. On Page 4, Line 21: PAN mixing ratios are on the order of several ppbv in heavily polluted air-masses. This is strange here.

Within the Russo et al (2003) paper, they look at polluted air masses from several regions. The reviewer is correct that in some regions, below 2 km in altitude, PAN concentrations, detected in aircraft campaigns and using backward trajectories, can reach several ppbv. However, in the majority of regions, the [PAN]s are less than 1.0 ppbv. In the mid-upper troposphere (>2 km in altitude), there are a few cases where max [PAN]s get to 900-1000 pptv. Therefore, we have altered the statement "rising to over 600 pptv in polluted air masses (Russo et al., 2003)." on page 4, line 21 to "rising up to 1000 pptv in some polluted air masses (Russo et al., 2003).".

[revised manuscript text omitted]

---

## Author Response (AR2)

**Editor Comment Responses:**

The Editor's comments are in black text and our responses are in red text:

Technical points:

\* in Eq. 2, do you mean A \cdot x on the right hand side of the equation?

This has been altered.

\* replace F11 by CFC-11 and F22 by HCFC-22

These have been changed.

\* Ref. Mogili: remove curly brackets in the title

This has been corrected.

\* I think journal names should be abbreviated

This has been changed.

\* The recent paper by Ungermann (see below) give some further information on PAN enhancements in the monsoon -- perhaps this is helpful. Please do not feel pushed to cite the paper!

Observations of PAN and its confinement in the Asian summer monsoon anticyclone in high spatial resolution, Jörn Ungermann, Mandfred Ern, Martin Kaufmann, Rolf Müller, Reinhold Spang, Felix Ploeger, Bärbel Vogel, and Martin Riese, Atmos. Chem. Phys., 16, 8389-8403, doi:10.5194/acp-16-8389-2016, 2016

We have added the following text on P5, L12 "More recently, Ungermann et al. (2016) used observations from the Cryogenic Infrared Spectrometers and Telescopes for the Atmosphere (CRISTA) infrared limb sounder, on-board the NASA Space Shuttle in August 1997, to investigate the enhancement of PAN in the Asian summer monsoon anticyclone. At 380 K, CRISTA retrieved peak PAN concentrations of over 350 pptv.".

[revised manuscript text omitted]